# Aspects to consider regarding breast cancer risk in trans men: A systematic review and risk management approach

Edvin Wahlström[1]*, Riccardo A. Audisio[2], Gennaro Selvaggi[1]

**1** Department of Plastic Surgery, Institute of Clinical Sciences, University of Gothenburg, Gothenburg, Sweden, **2** Department of General Surgery, Institute of Clinical Sciences, University of Gothenburg, Gothenburg, Sweden

☉ These authors contributed equally to this work.
* Edvin.wahlstrom@gu.se

## Abstract

### Background

The risk of breast cancer in trans men is currently a poorly understood subject and trans men likely carries a different level of risk from that of cis women.

### Aim

This review aims to review several aspects that affects breast cancer risk in trans men and to apply the Swiss cheese model to highlight these risks. The study takes its cue from a systematic review of all described breast cancer cases in trans men following medical or surgical intervention because of gender dysphoria.

### Methods

PubMed was systematically searched on the 14th of March 2023 to find all published cases of breast cancer following chest contouring surgery in trans men. Included articles had to involve trans men, the diagnosis of breast cancer had to be preceded by either a medical or surgical intervention related to gender dysphoria, and cases needed to involve invasive breast cancer or ductal carcinoma in situ. Articles were excluded if gender identity in the case subject was unclear and/or a full English version of the report was unavailable. Quality and risk of bias was evaluated using the GRADE protocol. A literature review of specific risk altering aspects in this population followed. The Swiss cheese model was employed to present a risk analysis and to propose ways of managing this risk.

### Results

28 cases of breast cancer in trans men have been published. The Swiss cheese model identified several weaknesses associated with methods of preventing breast cancer in trans men.

**Data Availability Statement:** All relevant data are within the manuscript and its Supporting Information files.

**Funding:** The author(s) received no specific funding for this work.

**Competing interests:** The authors have declared that no competing interests exist.

### Clinical implications

This study may highlight the difficulties with managing risk factors concerning breast cancer in trans men to clinicians not encountering this patient group frequently.

### Conclusion

This review finds that evidence for most aspects concerning breast cancer in trans men are inadequate, which supports the establishment of a risk-management approach to breast cancer in trans men.

## Introduction

Breast cancer is the most common form of cancer in the biological female population and constitutes almost 25% of all new female cancers [1]. The Global Cancer Observatory estimates that 47.8 in 100,000 cis women are diagnosed with breast cancer annually [2]. For the Swedish population the highest incidence is found in women aged 70 to 74 years [3].

People with gender dysphoria (GD) likely carry different risk of breast cancer compared to cis women. The risk of breast cancer is probably affected by both the surgical and medical treatments offered to these patients. The current knowledge concerning the risk altering effects of these procedures, in both the pre- and postoperative trans male population, for breast cancer remain unknown.

Other unknown aspects, specific to the trans male population, might also alter the breast cancer risk profile. These issues suggest that breast cancer in the trans male population could develop into a public health issue and, as such, should be regulated by policy and guidelines. In this review, we searched the available literature to determine the epidemiology of breast cancer in trans men both pre- and post-chest contouring mastectomy (CCM). Additionally, we discuss the rationale for breast cancer screening in trans men and whether histopathologic examination of removed breast tissue should be required at the time of CCM.

## Materials and methods

### Search methodology

We performed a systematic literature search on the 14[th] of March 2023, using PubMed to identify all published cases of breast cancer in trans men. Eligibility for inclusion was as follows: (1) cases had to involve trans men, (2) breast cancer diagnosis needed to be preceded by a GD-related intervention (either androgen therapy or a type of CCM), and/or (3) cases needed to involve invasive breast cancer or ductal carcinoma *in situ*. Exclusion criteria were as follows: (1) gender identity in the case subject was unclear and/or (2) a full English version of the report was unavailable.

The first author (EW) screened all articles produced by the final search first by titles, followed by abstracts and finally full-text. All titles excluded after full-text review and the reasonings for it can be found in S1 Table. EW retrieved all data gathered from the included articles and synthesized these as presented in the results section.

Search terms were derived from relevant articles either previously known to the authors or through an initial unstructured search. These initial search terms were then divided into subject and object categories and complemented with corresponding Medical Subject Heading terms. Synonyms and conjugations of the search terms were added to the final search, which

was then validated by cross-referencing all articles used to derive the initial search terms. The final search terms were as follows: ("breast cancer"[Text Word] OR "breast malignancy"[Text Word] OR "breast Neoplasms"[Text Word] OR "Breast carcinoma"[Text Word] OR ("breast"[Text Word] AND "carcinoma"[Text Word]) OR ("breast Neoplasms"[MeSH Terms] OR "carcinoma, ductal, breast"[MeSH Terms] OR "breast carcinoma in situ"[MeSH Terms])) AND ("transgender"[Text Word] OR "transgender male*"[Text Word] OR "female-to-male"[Text Word] OR "female-to-male"[Text Word] OR "transexual*"[Text Word] OR "trans man"[Text Word] OR "trans men"[Text Word] OR ("transgender persons*"[MeSH Terms] OR "transsexualism*"[MeSH Terms] OR "gender identity"[MeSH Terms])). The full search methodology can be found in S2 Table.

Additionally, we searched PubMed, Scopus, and Google Scholar to identify studies on breast cancer screening in the trans male population, as well as those reporting existing indications for histopathologic examination following GD-related mastectomy.

No study protocol was established, and the review have not been registered.

### Risk analysis

To illustrate ways of mitigating the risk of breast cancer in trans men we employed the Swiss cheese model of risk analysis [4]. The model illustrates how a hazard can develop into an accident. It assumes accidents often are the outcome of a series of failures. The model offers a structured way of identifying protective factors reducing hazards, and what weaknesses in these factors risk leading to accidents. We used the model to illustrate the hazard of breast cancer in trans men, what specific factors that could work protectively, and what weaknesses in these protective factors could be identified.

### Structural design

The search method was illustrated using the PRISMA flowchart template [5]. All studies used in this systematic review were assessed for quality by EW using the Grading of Recommendations, Assessment, Development, and Evaluations protocol [6]. No statistical synthesis of results were conducted from the included studies.

### Ethics

This review only reports on previously published data. As such, approval from an ethical board has not been indicated.

## Results

### Cases of breast cancer in trans men

Database searches yielded 374 results. Fig 1 describes the screening process. After screening titles and abstracts, 51 articles were selected for full-text screening. We identified and included five additional articles from the references of the 51 articles, resulting in a total of 56 articles. Of these, 34 were removed due to inclusion/exclusion criteria. A complete list of reasons for the exclusions is provided in S1 Table. Of the 22 included articles, there were five retrospective observational studies [7–11] and 17 case reports [12–28]. Table 1 describes all included cases.

Among the 22 articles, we identified 28 unique cases, with nine of these describing findings that occurred after CCM [9, 13, 14, 19–21, 24]. Time to breast cancer diagnosis after CCM was specified in six cases [13, 14, 19–21, 24] and ranged from 1 to 20 years (median: 8.5 years). All 28 cases received cross sex hormone treatment (CSH) preceding breast cancer diagnosis, although one case had terminated CSH treatment 10 years prior to diagnosis and after 10 years of treatment. The mean duration of CSH to breast cancer diagnosis was 6.7 years (median: 3.5

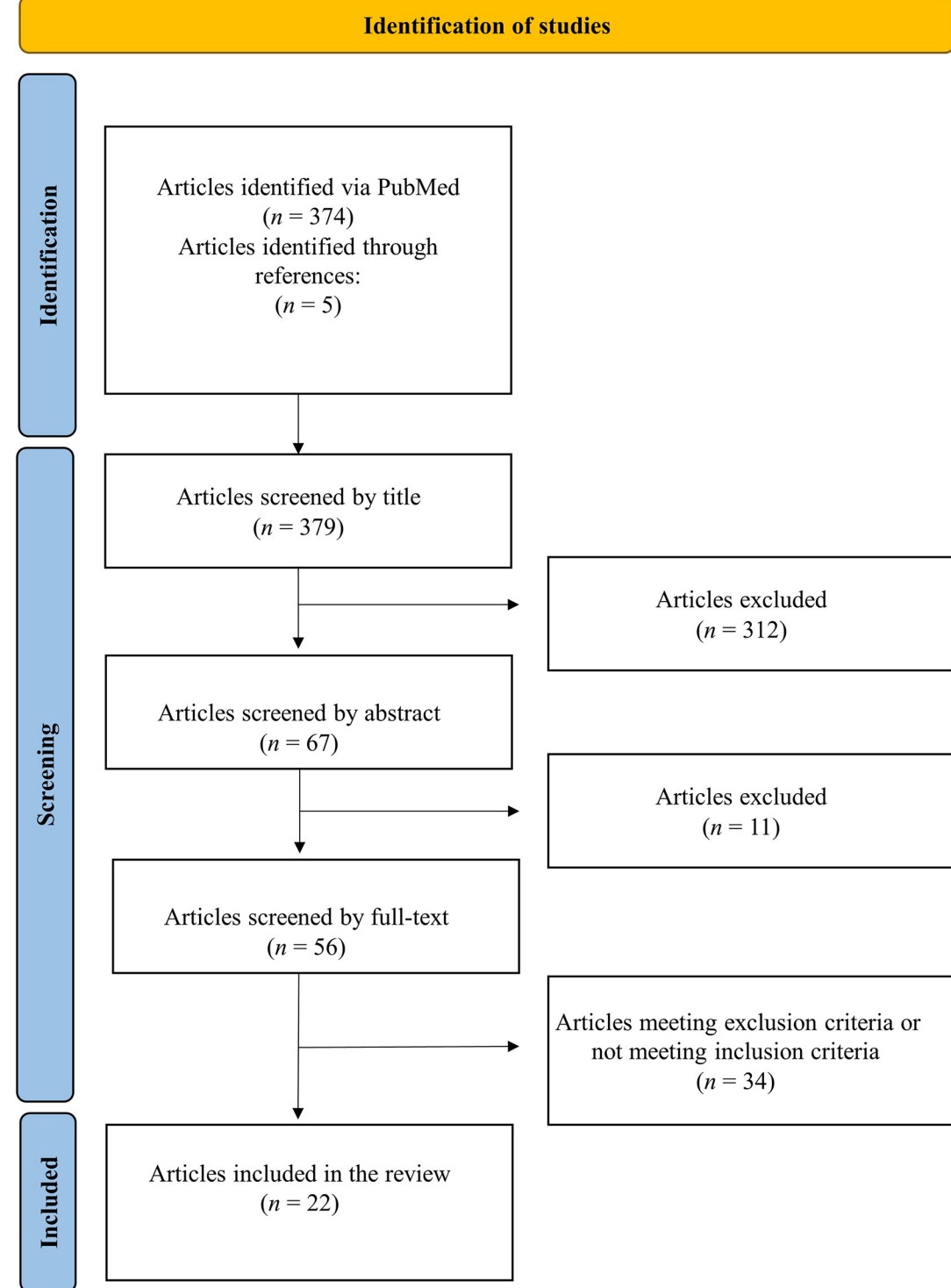

**Fig 1. PRISMA flowchart.** This flowchart describes the screening and inclusion process.

years). Age at breast cancer diagnosis was specified in 24 cases (mean: 41 years, 10 months) [7, 8, 10–28]. In the other four cases, age at diagnosis reportedly varied from 35 to 59 years (median: 47 years) [9].

**Table 1. Reported cases of breast cancer in trans men.**

| Study | Age at diagnosis (y) | Breast cancer type | ER | PR | AR | HER2 | Testosterone treatment prior to diagnosis | Mastectomy prior to diagnosis | BRCA1/2 |
|---|---|---|---|---|---|---|---|---|---|
| Baker et al. (2021) [7] | 29 | DCIS | NR | NR | NR | NR | 61 mo | Incidental finding at CCM | NR |
| Brown & Jones (2015) [8][a] | 78 | NR | + | – | NR | NR | 11 y | NR | NR |
| de Blok et al. (2019) [9] | NR[b] | NR | NR | NR | NR | NR | Yes[c] | Yes[c] | NR |
| de Blok et al. (2019) [9] | NR[b] | NR | NR | NR | NR | NR | Yes[c] | Yes[c] | NR |
| de Blok et al. (2019) [9] | NR[b] | NR | NR | NR | NR | NR | Yes[c] | Yes[c] | NR |
| de Blok et al. (2019) [9] | NR[b] | NR | NR | NR | NR | NR | Yes[c] | Incidental finding at CCM | NR |
| Gooren et al. (2013) [10] | 27 | Grade 1 tubular adenocarcinoma | + | + | NR | NR | 3 y | Incidental finding at CCM | NR |
| van Renterghem et al. (2018) [11] | 31 | Invasive carcinoma | + | + | NR | – | 16 mo | Incidental find at CCM | NR |
| Barghouthi et al. (2018) [12] | 28 | Grade 3 invasive ductal carcinoma | – | – | – | + | 1 y | No | – |
| Burcombe et al. (2003) [13] | 33 | Grade 1 ductal carcinoma (pT4aN0M0) | + | + | NR | NR | 13 y | Yes, 10 y before diagnosis | NR |
| Chotai et al. (2019) [14] | 58 | Grade 3 invasive carcinoma with ductal features | + | + | NR | NR | 10 y[d] | Yes, 20 y before diagnosis | NR |
| Eismann et al. (2019) [15] | 29 | DCIS | + | NR | NR | NR | 4 y | Incidental finding at CCM | – |
| Fehl et al. (2019) [16] | 41 | Invasive ductal carcinoma | + | + | + | + | 7 mo | Palpable mass detected before CCM | NR |
| Fledderus et al. (2020) [17] | 50 | DCIS | NR | NR | NR | NR | 3 y | Incidental finding at CCM | NR |
| Fundytus et al. (2020) [18] | 48 | Invasive ductal carcinoma stage IIA (pT1pN1M0) | + | + | + | – | 19 y | Partial mastectomy, later incidental finding at CCM | NR |
| Gooren et al. (2015) [19] | 48 | Infiltrating ductal carcinoma | – | – | NR | – | 9 y | Yes, 7 y before diagnosis | NR |
| Gooren et al. (2015) [19] | 41 | Tubular adenocarcinoma (pT1aN0M0); TNM stage I. | + | + | NR | – | 1 y | Incidental finding at CCM | NR |
| Katayama et al. (2016) [20] | 41 | Neuroendocrine carcinoma (pT1cN0M0) | + | + | + | NR | 15 y | Yes, 12 y before diagnosis | NR |
| Kopetti et al. (2021) [21] | 28 | Invasive breast carcinoma (pT2cN0cM0) | + | – | + | + | 30 mo | Yes, 2 y before diagnosis | – |
| Light et al. (2020) [22] | 44 | Invasive ductal carcinoma | + | + | + | – | 3 mo | Incidental finding before CCM | NR |
| Mingrino & Wang (2021) [23] | 51 | DCIS | – | NR | + | + | 1 y | Incidental finding at CCM | – |
| Nikolic et al. (2012) [24] | 42 | Invasive ductal carcinoma T2N2M1 | – | – | + | + | 30 mo | Yes, 1 y before diagnosis | NR |
| Parmeshwar et al. (2021) [25] | 31 | Invasive ductal carcinoma | + | + | + | – | 17 y | Incidental find at physical examination prior to CCM | – |
| Shao et al. (2011) [26] | 53 | Grade 2 invasive ductal carcinoma | + | – | NR | + | 5 y | No | – |
| Shao et al. (2011) [26] | 27 | Invasive ductal carcinoma | + | + | NR | + | 6 y | No | – |
| Tanini et al. (2019) [27] | 36 | Invasive carcinoma | + | + | + | + | 3 y | No | – |

*(Continued)*

**Table 1.** (Continued)

| Study | Age at diagnosis (y) | Breast cancer type | ER | PR | AR | HER2 | Testosterone treatment prior to diagnosis | Mastectomy prior to diagnosis | BRCA1/2 |
|---|---|---|---|---|---|---|---|---|---|
| **Tanini et al. (2019) [27]** | 33 | Grade 3 DCIS | + | + | + | NR | 30 mo | Incidental finding at CCM | NR |
| **Treskova et al. (2018) [28]** | 58 | Invasive ductal carcinoma | + | + | NR | + | 25 y | Incidental finding during preoperative screening for CCM | NR |

This table shows all reported cases of breast cancer in trans men after receiving either cross sex hormone treatment or undergoing chest contouring mastectomy. Footnotes.

[a] This was the only case in the article reporting androgen treatment prior to diagnosis.

[b] Age at breast cancer diagnosis varied between ages 35–59 (median 47).

[c] All patients received testosterone treatment for a median of 15 years (range: 2–17 years) prior to diagnosis was 15 years (range 2–17). Three of four cases were diagnosed several years after CCM.

[d] Received testosterone injections for 10 years but stopped thereafter on his own accord 10 years before diagnosis.

CCM = chest contouring mastectomy, NR = not reported, ER = estrogen receptor, PR = progesterone receptor, AR = androgen receptor, DCIS = ductal carcinoma in situ, HER2 = human epidermal growth factor receptor 2,"–"= Negative, "+" = Positive.

## Quality assessment of the included studies on breast cancer in trans men

All but one study were of very low quality [7, 8, 10–28] and demonstrated poor study design. All case reports were downgraded due to bias derived from the study design and publication bias. Of the five observational studies, four were assessed as very low quality [7, 8, 10, 11] for the following reasons: (1) improper reporting of confounding factors [8, 10], (2) the study was underpowered [7, 8, 11], and (3) the study was improperly controlled [7, 10, 11].

We graded one retrospective observational study as low quality [9]. This was not downgraded further for the following reasons: (1) the study was adequately powered; and (2) although breast tissue from trans men more often underwent histopathologic examination relative to that from cis women, the study still found a lower incidence of breast cancer in the evaluated trans men relative to the evaluated cis women. Table 2 describes the quality assessment.

## Epidemiology of breast cancer in trans men

Gooren et al. reported a breast cancer incidence of 5.9 in 100,000 trans men in a Dutch cohort [10]. Recently, De Blok et al., also in a dutch cohort, calculated an incidence ratio of 0.2 for trans men as compared with cis women and 58.9 for trans men to cis men, also noting that within the Dutch population, breast cancer in trans men is diagnosed at an earlier age (median: 47 years; range 35–59 years) relative to cis women (average: 61 years) [9]. The age at breast cancer diagnosis in the trans male population is consistent with the age ranges reported in the 28 cases from the included articles. Notably, four of the 28 cases included in this review involved the cohort evaluated by de Blok et al.

Among studies estimating breast cancer incidence in trans men, the overall age of the cohorts is often relatively young: the median age of de Block et al.'s cohort was 39 (interquartile range 26–51) [9]. Gooren et al. (2013) reported a mean age of 23.2 ± 6.5 years of their cohort, with a median follow-up of 16.8 years (range 6.0–36 years) [10]. Despite the follow-up period, the study did not evaluate higher age groups, which typically present higher incidences of breast cancer across the population of cis women [29].

Brown and Jones have presented the oldest cohort to date when investigating the breast cancer incidence in a cohort of 5,135 transgender patients. Of these, 1,579 were assigned female sex at birth and had a mean age of 55.6 years with 25% of patients above 65 years old.

**Table 2.  Quality assessment.**

| Article | Study design | Risk of bias | Inconsistency | Indirectness | Imprecision | Publication bias | Large effect | Dose response | Confounders reducing effect | GRADE |
|---|---|---|---|---|---|---|---|---|---|---|
| **Baker et al. (2021) [7]** | Observational | Uneven study populations. | — | No | Not adequately powered | Unlikely | | Cannot be determined. | No | Very low. |
| **Brown & Jones (2015) [8]** | Observational | Confounders not adequately reported. Study population not adequately reported. | — | No | Large CI. Not adequately powered. | — | — | | Not adequately reported. A possible major confounder is the histopathologic examination at surgery | Very low. |
| **de Blok et al. (2019) [9]** | Observational | Confounders not adequately reported in control group | — | No | Large CI across a 30–50-y timeline. Few events, although adequately powered | — | — | Inverted | Not adequately reported. A possible major confounder is histopathologic examination at surgery | Low. |
| **Gooren et al. (2013) [10]** | Obeservational | No control group. Confounders not adequately reported. | — | No | Few events, possibly fragile results. | — | | Cannot be determined. | Cannot be determined. | Very low. |
| **van Renterghem et al. (2018) [11]** | Observational | No control group and underpowered. | — | No | No | No | | Cannot be determined | No | Very low. |
| **Barghouthi et al. (2018) [12]** | Case report | High because of study design | — | No | — | Likely | — | — | — | Very low. |
| **Burcombe et al. (2003) [13]** | Case report | High because of study design | — | No | — | Likely | — | — | — | Very low. |
| **Chotai et al. (2019) [14]** | Case report | High because of study design | — | No | — | Likely | — | — | — | Very low. |
| **Eismann et al. (2019) [15]** | Case report | High because of study design | — | No | — | Likely | — | — | — | Very low. |
| **Fehl et al. (2019) [16]** | Case report | High because of study design | — | No | — | Likely | — | — | — | Very low. |
| **Fledderus et al. (2020) [17]** | Case report | High because of study design | — | No | — | Likely | — | — | — | Very low. |
| **Fundytus et al. (2020) [18]** | Case report | High because of study design | — | No | — | Likely | — | — | — | Very low. |
| **Gooren et al. (2015) [19]** | Case report | High because of study design | — | No | — | Likely | — | — | — | Very low. |
| **Katayama et al. (2016) [20]** | Case report | High because of study design. | — | No | — | Likely | — | — | — | Very low. |
| **Kopetti et al. (2021) [21]** | Case report/ review | High because of study design | — | No | — | Likely | — | — | — | Very low. |
| **Light et al. (2020) [22]** | Case report | High because of study design | — | No | — | Likely | — | — | — | Very low. |
| **Mingrino & Wang (2021) [23]** | Case report | High, because of study design | — | No | — | Likely | — | — | — | Very low. |

*(Continued)*

**Table 2.** (Continued)

| Article | Study design | Risk of bias | Inconsistency | Indirectness | Imprecision | Publication bias | Large effect | Dose response | Confounders reducing effect | GRADE |
|---|---|---|---|---|---|---|---|---|---|---|
| Nikolic et al. (2012) [24] | Case report | High because of study design | — | No | — | Likely | — | — | — | Very low. |
| Parmeshwar et al. (2021) [25] | Case report | High because of study design | — | No | — | Likely | — | — | — | Very low. |
| Shao et al. (2011) [26] | Case report | High because of study design | — | No | — | Likely | — | — | — | Very low. |
| Tanini et al. (2019) [27] | Case report | High because of study design | — | No | — | Likely | — | — | — | Very low. |
| Treskova et al. (2018) [28] | Case report | High because of study design | — | No | — | Likely | — | — | — | Very low. |

Quality assessment results for the included studies using the GRADE protocol.

Footnotes.

CI = confidence interval, GRADE = Grading of Recommendations, Assessment, Development, and Evaluations [6], "—"= Not applicable.

The authors reported a standardized breast cancer incidence ratio of 1.64 as compared with cis women, although statistically insignificant with CI 95% 0.24–7.22 [8]. If constrained to cases with prior testosterone use the reported standardized incidence ratio was 0,26 CI 95% 0–3.69, also statistically insignificant. Notably, this study did not exclusively evaluate trans men but rather included patients of "unspecified gender disorder" and "transvestic fetishism" in their cohort. Furthermore, they presented data from transgender men labelled as "females" and compared them with cis women also labelled as "females".

Previous systematic reviews of studies on breast cancer incidence in trans men also reported that the overall quality of studies was low [30, 31], and that the risk of bias in cohort studies is high [17]. One review concluded that cancer incidence in older trans men cannot be determined, and that breast cancer in trans men is diagnosed earlier as compared with that in cis women and cis men [32]. Detection of breast cancer in trans men at a younger age than cis women could be a result of age bias related to the rising incidence of GD accompanied by a lack of long-term follow-up in an adequate number of trans men. Further investigations are needed to clarify these findings.

## Breast cancer screening in cis women

The World Health Organization recommends that all women aged 50 to 69 years undergo a mammography every 2 years [33]. Current Swedish guidelines for cis women suggest a more extensive screening scheme, with routine mammography starting at age 40 and then repeating every 18 to 24 months until age 74. High-risk patients, such as *BRCA1/2* carriers, are offered magnetic resonance imaging (MRI) screening from age 25 to 55 years [34]. The increased sensitivity of MRI is used for special cases, with ultrasound recommended for those with an increased lifetime risk of breast cancer (at least a 20% likelihood), presenting dense breast tissue, and aged <50 years [35].

## Breast cancer screening in trans men

Specific guidelines for breast cancer screening in trans men before CCM mimic current recommendations for cis women [17, 36–46]. Roznovjak et al. [47] and Meggetto et al. [48]

**Table 3. Guidelines for breast cancer screening, pre- and post-chest contouring mastectomy, in trans men.**

| Source of guidelines | Pre-CCM | Post-CCM |
|---|---|---|
| *Fledderus et al. (2020) [17]* | Same screening as that used for cis females | Individual assessment |
| *Guidelines for gender-affirming primary care with trans and non-binary patients [36]* | Same screening as that used for cis females | No screening necessary. Breast self-examination should be encouraged. |
| *University of California, San Francisco Transgender Care [37]* | Same screening as that used for cis females | Dialogue between patient and physician |
| *Transgender Primary Medical Care: Suggested Guidelines for Clinicians in British Columbia (Canada) [38]* | Same screening as that used for cis females | Annual chest-wall and axillary exams and patient education on residual breast tissue |
| *The Fenway Institute [39]* | Same screening as that used for cis females | Annual chest-wall and axillary exams; educating patients about residual breast tissue and breast cancer risk |
| *University Hospitals, Ohio (United States) [40]* | Same screening as that used for cis females | Dialogue between patient and physician regarding appropriate screening modality; screening not specified |
| *NHS Scotland [41]* | Same screening as that used for cis females | No screening necessary |
| *Susan G. Komen organization [42]* | Same screening as that used for cis females | Annual chest-wall and axillary exams |
| *Cancer Care Manitoba, Canada [43]* | Same screening as that used for cis females | Dialogue between patient and physician |
| *Canadian Cancer Society [44]* | Same screening as that used for cis females | Same screening as that used for cis females |
| *Cancer Research UK [45]* | Same screening as that used for cis females | No screening; mammograms not considered feasible |
| *Breastscreen Victoria [46]* | Same screening as that used for cis females | Dialogue between patient and physician |

This table demonstrates the heterogeneity of recommendations for screening protocols following chest contouring mastectomy in trans men.

Footnotes.

CCM = chest contouring mastectomy.

reported a lack of consistency in existing recommendations for screening in trans men following CCM. For example, some hospitals in the United States advise discussion between physician and patient regarding screening. Additionally, the Susan G. Komen organization recommends annual chest and axillary exams by a health care professional [42], and The Canadian Cancer Society recommends mammograms every 2 years between age 50 and 69 years [44]. Table 3 summarizes the current guidelines.

## Histopathologic examination of breast tissue from cis women

Similar to CCM, reduction mammoplasty (RM) is routinely performed on breast tissue that is assumed to be free of malignancies. Routine histopathologic examination of excised breast tissue following RM, independent of age, is common in the field of plastic surgery [49, 50].

Importantly, the cost of histopathologic examination when accounting for the rarity of malignant findings in patients <40 years of age makes this decision one of cost versus benefit [50].

Several authors offer different recommendations regarding when histopathology following RM is indicated. Ambaye et al. [51] evaluated breast tissue samples collected from RMs in 595 patients, concluding that histologic examination is only necessary in patients aged <35 years when clinical lesions are present or in cases of a strong family history of breast cancer. For patients aged >35 years, they recommended gross examination of breast tissue by a pathologist, followed by histopathologic examination of up to seven breast sections or up to six sections for patients aged >50 years [51]. Analysis of the pathology records of 1,388 RM patients by Hassan and Pacifico [52] led to a recommendation of histopathologic assessment starting at age 30. Conversely, Merkkola-von Schantz et al. [53] suggested that all post-RM specimens should undergo pathologic assessment.

The Royal Collage of Pathologists deems routine histopathologic examination following RM to be of limited value [54]. The American Society of Plastic Surgeons recommends histopathology "when clinically indicated and after careful consideration of patient history, risks, and benefits" [55]. Furthermore, Swedish national guidelines recommend that RM specimens undergo histopathologic examination for patients aged ≥40 years, except in cases of a family history of breast cancer or the presence of genetic mutations (e.g., *BRCA1/2*) [56].

Epidemiological data should also be considered, given the varying prevalence of breast cancer between countries. According to the International Agency for Research on Cancer, the age-standardized breast cancer incidence rate in Sweden is estimated at 83.9 in 100,000 cis women, whereas Belgium presents the highest incidence in the world at 113.2 in 100,000 cis women (the worldwide incidence is estimated at 47.8 in 100,000 cis women) [2]. Therefore, guidelines for histopathologic examination of breast tissue likely need to be adjusted according to incidence based on location. This was emphasized in Hassan and Pacifico´s article by a statement that their results (and thus their recommendations) were best compared/applied to geographically similar populations [52].

## Histopathologic examination of breast tissue from trans men

To date, no guidelines regarding histopathologic examinations after CCM have been established. Despite the lack of evidence, recommendations for histopathologic examination following CCM have been proposed. Two recent articles recommend that all excised breast tissue undergo pathologic examination [57, 58], with Hernandez et al. (2020) recommending examination of four tissue blocks per mastectomy [58]. At our clinic, histologic examination of excised breast tissue is routinely performed for all trans men following CCM; however, there are no studies confirming this practice as a clinical necessity.

## Comparison and applicability

Similarities exist between RM in cis women and CCM in trans men. First, both patient groups present similar variance in family history of breast cancer, which can affect preoperative breast radiological or genetic screening. Moreover, this outcome could necessitate deviation from the original desired procedure (RM for cis women and CCM for trans men) and result in patient referral to a breast oncoplastic specialist. Second, when presenting a negative family history of breast cancer, both patient groups will undergo a type of procedure that includes breast remodelling. This involves reductions in breast glandular tissue along with retention of an unspecified amount of this tissue. Given these similarities, existing guidelines for RM could highlight the benefit of histopathologic examination after CCM in trans men.

Notably, there are also differences between RM and CCM patient groups. First, cross-sex hormone treatment (CSH) is an obvious and questionable difference, although there is no consensus on whether and to what extent CSH affects the risk of developing breast cancer. Second, as previously noted, it is possible that some trans men may not adhere to established guidelines for breast cancer screening. A study performed in a primary care setting in Canada reported lower screening rates for transgender patients as compared with cis gendered patients [59]. Third, we hypothesize that trans men likely demonstrate a decreased willingness to follow consistent self-examination of the breasts, given that an aversion toward female secondary sex characteristics is common in patients with GD.

Existing breast cancer screening guidelines for trans men pre-CCM mirror current guidelines for cis women (Table 2). As a result, the expectation should be that post-CCM histopathologic examination for trans men adheres to guidelines for post-RM histopathologic examinations for cis women. Given the relatively young age of trans men diagnosed with breast cancer, it remains to be determined whether a more cautious approach to both screening guidelines and the necessity for histopathologic examination following CCM are considered. Moreover, decisions need to be made as to whether these guidelines should be adjusted based on the incidence of breast cancer for a specific population. Furthermore, it remains to be determined whether the young age of trans men presenting with breast cancer is a consequence of age bias due to the increasing number of young trans men requesting a mastectomy.

## Surgical considerations regarding breast cancer risk

Breast glandular tissue that is retained following a mastectomy in trans men is usually more abundant than that retained in cis women. Despite this, physicians may assume that trans men post-CCM would be left with the same amount of breast tissue as cis women following prophylactic mastectomy. CCM is typically not a radical procedure, with some breast tissue retained to allow reconstruction of the chest to provide a masculine appearance. Specifically, breast tissue is retained at the level of the axilla, if present, and the superior quadrants by some surgeons in order to maintain resemblance of the pectoralis major muscles and/or anatomic symmetry with the rest of the upper body in patients presenting a high body mass index [60]. The nipple areola complex and some of the tissue underneath is retained in nearly every case in order to maintain vascularization [61].

A survey of plastic surgeons performing CCM in the United States confirmed that only 28% of respondents routinely remove all of the breast tissue during CCM [62], confirming that CCM is not comparable to prophylactic mastectomies and, therefore, not relevant to eliminating the risk of breast cancer. However, a previous study indicated the likelihood that reducing the amount of breast tissue lowers the risk of developing breast cancer. In a retrospective review of 31,910 medical records of Swedish patients that underwent breast reduction, Boice et al. reported an incidence ratio for breast cancer of 0.72 as compared with the general population [63].

## Cross sex hormone therapy

Assumptions of a low risk of breast cancer following CCM in trans men are based on the small amount of retained breast tissue and low levels of estrogen [20]. However, high levels of endogenous testosterone have been linked to an increased risk of breast cancer in cis women [64–68]. In trans men, high doses of CSH are administered and result in testosterone serum levels higher than those in cis women and similar to those in cis men [69]. The impact of these levels on breast cancer risk, as well as their cancer-specific mechanisms, in trans men remains unknown.

In cis women that produce high levels of testosterone, studies suggest that increased aromatase activity involved in estrogen production represents a possible oncogenic pathway [64]; however, the presence of a similar pathway in trans men remains to be determined. Additionally, a retrospective chart review by Chan et al. [70] with a 6-year follow-up showed low estradiol levels and no increases in these levels in CSH-treated trans men. This suggests that peripherally converted testosterone should not be an issue in trans men. Although not measured in their investigation, Chan et al. advised that intracellular conversion might still occur. Under such conditions, CSH might increase the risk for breast cancer in trans men through aromatase conversion. Furthermore, pathways related to androgen receptors could potentially be involved in breast cancer risk in trans men based on their roles in breast cancer in cis women [65, 66].

## Breast cancer screening modalities for trans men

There are no reports on screening modalities for breast cancer in trans men, making it unclear to what extent mammography sensitivity and specificity exist pre- and post-CCM. There are numerous potential differences in screening modalities between cis women, trans men, and cis men.

First, it remains to be determined whether anatomical differences created by the choice of CCM technique affect breast screening. Specifically, it is unknown whether special radiological techniques in addition to or instead of mammography are necessary to perform an equally effective assessment post-CCM according to anatomical changes. Given the similarity in the gross anatomy of the chest between cis men and post-CCM trans men, it is likely that guidance on screening modalities in cis men can be effectively utilized.

A recent study that included high-risk cis males that underwent mammogram screening reported 100% sensitivity and 95% specificity for that method [71], which agrees with its use as the primary imaging modality recommended for cis men [72]. This suggests that mammography post-CCM would likely be equally effective in trans men, although the anatomical effects of CCM on these outcomes remain to be determined. Importantly, a previous study showed that scarring after RM can increase the complexity of interpreting mammograms, suggesting complementary use of a secondary modality for the first screening post-RM [73].

Second, it is unclear how CSH therapy affects breast tissue and screening performance pre- and post-CCM. Histopathology studies report an increased ratio of fibrous stroma in trans men receiving CSH [74–77]. Recently, Baker et al. showed that trans men receiving CSH presented more extensive lobular atrophy; fewer cysts, fibroadenomas, and papillomas; and decreased pseudoangiomatous stromal hyperplasia and inflammation relative to those not receiving CSH [7]. However, the authors were unable to show significant differences between groups regarding atypical lesions, such as ductal carcinoma or atypical ductal hyperplasia.

Additionally, Torous and Schnitt identified calcifications in 22% of the examined breast tissue from trans men as compared with 6% in breast tissue from cis women who had undergone breast-reduction [74]. Given that calcifications can be a sign of malignancy [78], an increased ratio of these in trans men could increase the number of biopsies required according to many screening programs.

Given the effects of both surgery and CSH on breast parenchyma, available screening modalities for trans men need to be validated, including their specificity, sensitivity, the number of exams leading to biopsies, and the risk/benefit ratio of mammography.

## Screening-related harm

The risk/benefit ratio is a central aspect of any screening program. A 2012 report on breast cancer screening programs in the United Kingdom concluded that overdiagnosis was the

major contributor to harm associated with screening outcomes [79]. The report estimated an ~11% chance of risk associated with overdiagnosis while acknowledging that such estimations are complicated by the different methodologies. Other harms in these cases include damage from radiation treatment, incidence of false-positives/-negatives, potential morbidity/mortality, and the psychological impact of the diagnosis.

When performing mammograms, the final dose of radiation varies according to the number and type of images collected. Although risk of radiation-induced cancer is low, the benefits greatly outweigh the risk in these cases [79]. A study by Berrington de Gonzalez estimated that only between three and six in 10,000 women would develop cancer as a result of screening-related radiation, assuming that they underwent screening every 3 years between the ages of 47 and 73 years [80].

### Risk management

Because the incidence of GD has increased in recent decades [81], we expect that the incidence of breast cancer in trans men will likely increase in proportion. As a result, the need for evidence-based guidelines, treatments, and prevention measures will also increase. There currently exist few studies on different aspects related to these issues in trans men. Although epidemiological studies have been conducted on trans men, few included cohorts at an advance age, thereby failing to offer insight into a lifetime risk of breast cancer in trans men. Given that the current status of research in this area suggests the likelihood of such risks being underestimated, a risk-management approach is appropriate.

Risk management aims to mitigate risks before mistakes/errors occur, making it particularly applicable in situations where the magnitude of outcomes is uncertain. For example, a recent study applied longitudinal risk management to patients with an increased risk of breast cancer [82]. The Swiss cheese model is a risk-management approach that categorizes poor outcomes as follows: (1) active failures resulting from actions by people in direct contact with patients or systems (i.e., procedural complications due to negligence or deviation from protocol) or (2) mistakes or errors resulting from latent conditions that represent underlying weaknesses in the system [4].

Active failures can be mitigated by standardizing procedures but are difficult to completely remove because of the inherent nature of humans toward making mistakes. By contrast, mitigating the risk of latent conditions is more easily sustained due to its systematic nature. Disease screening represents a method for mitigating the risk of a latent condition. Mitigating active failures in this context could involve increasing the awareness of a screening program and encouraging patient participation.

The Swiss cheese model can identify the barriers that exist to decreasing the risk of something happening and highlight weaknesses that allow mistakes to happen. To apply this model to manage the risk of breast cancer in trans men, we use cheese slices to represent preventive measures taken by health care entities and holes in the cheese slices to represent the weaknesses of the system (Fig 2).

Illustration of the five barriers identified as part of the Swiss cheese model. To apply this model to manage the risk of breast cancer in trans men, we use cheese slices to represent preventive measures taken by health care entities and holes in the cheese slices to represent the weaknesses of the system.

### Slice 1: Identify high-risk patients

This includes a review of patient medical and family history in order to identify risk status in relation to breast cancer and in consideration of CCM planning. Therefore, holes in Slice 1

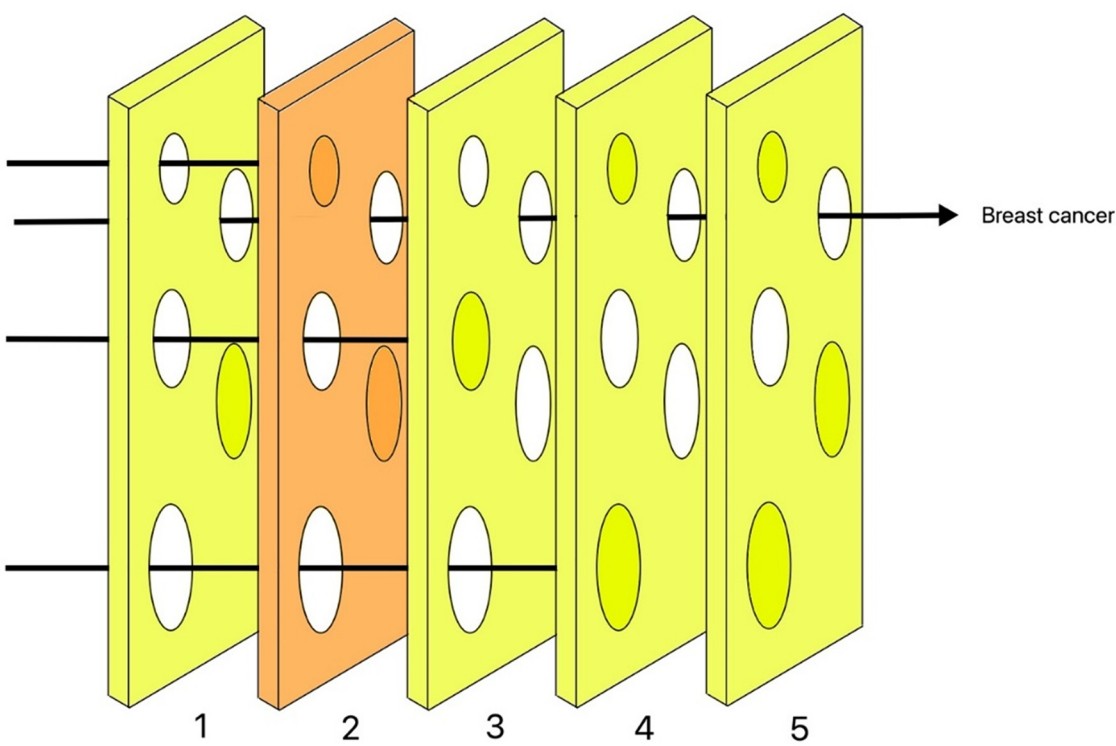

**Fig 2. Swiss cheese model.**

include not identifying patients with a genetic predisposition to cancer and failing to offer prophylactic mastectomy.

### Slice 2: Screening

This includes employment of pre-surgery mammograms and breast cancer screening guidelines. Additionally, this would include the use of personnel trained in caring for the subset of trans men indicated for breast cancer screening (patients aged >40 years or at high risk). Holes for Slice 2 include low participation in screening programs, perceived discrimination by health care workers, aversion to examinations perceived as a female activity, improperly performed mammograms, and exclusionary wording used in screening materials.

### Slice 3: Education

This includes making patients aware that residual breast tissue remains intact following CCM, and that breast cancer continues to be a possibility. Additionally, physicians may need to be educated regarding CCM not being equivalent to prophylactic mastectomy. Holes for Slice 3 include a delay in seeking health care or screening opportunities due to a lack of knowledge regarding the continued cancer risk. Similarly, delayed diagnosis can occur in cases of physicians being unaware of the existence of residual breast tissue.

### Slice 4: Participation in post-surgery follow-up visits

Post-surgery follow-up is important for both patients and health care professionals to facilitate a long-term relationship, especially given the lack of knowledge about breast cancer risk in trans men. This would increase the likelihood of early detection of symptoms, as well as

facilitate data collection for epidemiological studies. Holes for Slice 4 include inadequate care that complicates both patient and physician education, thereby hindering a cohesive understanding of how patients experience treatments. Holes in both Slices 2 and 4 risk patients losing their opportunity to be screened, if needed.

### Slice 5: Histopathologic examination of excised breast tissue

Given the lack of evidence-based guidelines for post-CCM trans men, guidelines used for RM in cis women should be employed until guidelines specifically for trans men have been developed. To maximize safety, all specimens should be examined when economically and practically feasible. Holes for Slice 5 include lower participation in breast cancer screening in trans men relative to cis women, which can result in late diagnosis of cancer.

## Discussion

Breast cancer in trans men continues to be an issue of unknown magnitude. Assessing the risk of breast cancer in trans men relative to that in cis women requires a deeper understanding of different disease specifications in trans men, including the age at which breast cancer develops, the impact of CSH on breast cancer risk, the effect and possible harm of screening (timing and modalities), and the utility of histopathologic examination at the time of CCM.

Because evidence-based screening guidelines for trans men have not been developed, those used for cis women are employed, with consensus on their application limited to the preoperative status of trans men. Guidelines for screening trans men postoperation vary considerably (Table 3). No studies on specific screening modalities aimed at trans men currently exist, and it remains uncertain whether mammography, ultrasound, or MRI is best suited for these patients. Furthermore, no studies have determined whether histopathologic examinations should be performed following mastectomies.

Mammography is a successful breast cancer screening and diagnostic method for cis men [71]. However, the physiological differences that exist between trans men and cis women suggest that it may not be appropriate to simply apply the same screening and diagnostic guidelines for breast cancer for both patient groups. To address this, epidemiological studies on older trans men and powered to establish a lifetime risk of breast cancer in trans men are needed to justify screening programs. Additionally, breast cancer risk in trans men carrying oncogenic mutations should be investigated in order to establish specific guidelines for these subgroups.

### Limitations

All studies included in this review are of low to very low quality, as such no certain conclusions regarding the epidemiology of breast cancer in trans men can be formulated. The systematic search in this review was limited in its scope to that of described breast cancer cases in trans men. Other aspects discussed have been reviewed thoroughly, but not in a structured manner. The study protocol was never published before the initiation of the study. Both of these limitations can introduce bias to the review. However, the impact of this bias towards the conclusions drawn in this review are considered to be limited since the conclusions are conservative, and a risk management approach is applied.

### Conclusions

The findings of this review support the establishment of a risk-management approach to breast cancer in trans men. Trans men should be educated about the presence of residual breast tissue

post-CCM and informed about the lack of evidence-based guidelines for breast cancer screening. Moreover, these patients should be offered a chance to participate in the decision-making process for breast cancer screening. Furthermore, based on previous findings, we suggest that histopathologic examinations should follow CCM in trans men as a breast cancer screening method. If this is not economically or practically feasible, the guidelines related to RM for cis women should be followed.

## Supporting information

**S1 Table. Excluded studies.** This table shows all studies excluded after full text review and states the reason behind exclusions. The inclusion criteria were as follows: (1) cases had to involve trans men, (2) breast cancer diagnosis needed to be preceded by a gender dysphoria-related intervention (either androgen therapy or a type of CCM), and/or (3) cases needed to involve invasive breast cancer or ductal carcinoma in situ. Exclusion criteria were as follows: (1) gender identity in the case subject was unclear and/or (2) a full English version of the report was unavailable.
(DOCX)

**S2 Table. Detailed searchterms.** This table describes the search process as conducted on PubMed. The search was constructed through 7 steps. The combined and final search were as follows: ("breast cancer"[Text Word] OR "breast malignancy"[Text Word] OR "breast Neoplasms"[Text Word] OR "Breast carcinoma"[Text Word] OR ("breast"[Text Word] AND "carcinoma"[Text Word]) OR ("breast Neoplasms"[MeSH Terms] OR "carcinoma, ductal, breast"[MeSH Terms] OR "breast carcinoma in situ"[MeSH Terms])) AND ("transgender"[Text Word] OR "transgender male*"[Text Word] OR "female-to-male"[Text Word] OR "female-to-male"[Text Word] OR "transexual*"[Text Word] OR "trans man"[Text Word] OR "trans men"[Text Word] OR ("transgender persons*"[MeSH Terms] OR "transsexualism*"[MeSH Terms] OR "gender identity"[MeSH Terms])). The search was conducted on the 14th of march 2023. It resulted in 374 hits.
(DOCX)

## Author Contributions

**Conceptualization:** Edvin Wahlström, Riccardo A. Audisio, Gennaro Selvaggi.

**Formal analysis:** Edvin Wahlström.

**Investigation:** Edvin Wahlström.

**Methodology:** Edvin Wahlström, Riccardo A. Audisio, Gennaro Selvaggi.

**Project administration:** Edvin Wahlström, Gennaro Selvaggi.

**Supervision:** Riccardo A. Audisio, Gennaro Selvaggi.

**Writing – original draft:** Edvin Wahlström, Riccardo A. Audisio, Gennaro Selvaggi.

**Writing – review & editing:** Edvin Wahlström, Riccardo A. Audisio, Gennaro Selvaggi.

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
