## [Decision Letter · Decision Letter 0]

8 Jan 2024

PONE-D-23-41281Aspects to consider regarding breast cancer risk in trans men: a literature review and risk management approach.PLOS ONE

Dear Dr. Wahlström,

Thank you for submitting your manuscript to PLOS ONE. After careful consideration, we feel that it has merit but does not fully meet PLOS ONE’s publication criteria as it currently stands. Therefore, we invite you to submit a revised version of the manuscript that addresses the points raised during the review process.

We look forward to receiving your revised manuscript.

Kind regards,

Daniele Ugo Tari, M.D.

Academic Editor

PLOS ONE

Journal Requirements:

2. Kindly add "systematic review" to the title of your manuscript

Reviewers' comments:

Reviewer's Responses to Questions

**Comments to the Author**

1. Is the manuscript technically sound, and do the data support the conclusions?

Reviewer #1: Yes

Reviewer #2: Yes

2. Has the statistical analysis been performed appropriately and rigorously? 

Reviewer #1: I Don't Know

Reviewer #2: Yes

3. Have the authors made all data underlying the findings in their manuscript fully available?

Reviewer #1: Yes

Reviewer #2: Yes

4. Is the manuscript presented in an intelligible fashion and written in standard English?

Reviewer #1: Yes

Reviewer #2: Yes

5. Review Comments to the Author

Reviewer #1: In general, this was an important review, although based on poor-quality research. There are a few corrections that need to be made to clarify the review.

- Beginning with the Abstract, the Swiss chess model is mentioned a number of times, but never explained at all until near the end in the Risk Management section. A concise explanation of the model is needed at the beginning. There are also two words in the Abstract that need correcting - in Methods, "the diagnosis of breast had" the word "cancer" needs to be added. Near the end of the same paragraph, "aspects in this populous" needs to be corrected to "population."

- In Materials and methods in Search methodology, "GD-related intervention" is used. Please define and spell out what GD stands for.

- In Comparison and applicability, in the second paragraph, "CSH is an obvious..." Please define and spell out CSH.

- In the Introduction, it is stated that the highest incidence of breast cancer is found in women aged 70 - 74 years. However, after checking the reference, this is very likely true only for Sweden; please verify that it is broadly applicable or clarify that this is only for Sweden.

- It is curious that only the first author screened all articles, read all text, and gathered all statistics. There was no double-check by the other authors (what exactly did they do?) and is a risk of bias. It is also curious that no protocol was established and the review has not been registered anywhere.

- Although the author is undoubtedly correct that there is more breast tissue remaining after CCM than after prophylactic mastectomy, it should still somewhat lower the risk of malignancy in trans men than in cis women at normal risk.

Reviewer #2: In the references section, this reviewer noticed two typographical errors:

Line 602, San Francisco is misspelled ("San fransisco" is incorrect)

Line 604, the second author on the cited document is Goldberg, J (not Joshua G)

6. PLOS authors have the option to publish the peer review history of their article (what does this mean?). If published, this will include your full peer review and any attached files.

Reviewer #1: No

Reviewer #2: No

---

## [Author Response · Author response to Decision Letter 0]

16 Jan 2024

Response to editor: 1: Format and style of the manuscript have been updated to adhere to guidelines. 

2: Systematic review was added to the title.

3: The reference list have been reviewed. Reference nr 69 was changed beacuse of a published erratum concerning the original citation.

Response to Reviewers

First, the authors would like to thank the reviewers for their time and effort in reviewing this manuscript. Many valid and thoughtful points have been raised. Our responses can be found beneath.

Reviewer #1: In general, this was an important review, although based on poor-quality research. There are a few corrections that need to be made to clarify the review.

- Beginning with the Abstract, the Swiss chess model is mentioned a number of times, but never explained at all until near the end in the Risk Management section. A concise explanation of the model is needed at the beginning. There are also two words in the Abstract that need correcting - in Methods, "the diagnosis of breast had" the word "cancer" needs to be added. Near the end of the same paragraph, "aspects in this populous" needs to be corrected to "population."

Response: A paragraph with the heading “Risk analysis” have been added to the materials and methods section explaining the swiss cheese model. The word cancer was added as suggested, populous were changed to population as well. 

#2 - In Materials and methods in Search methodology, "GD-related intervention" is used. Please define and spell out what GD stands for.

Response: GD is defined as gender dysphoria in the second paragraph of the introduction.

#3 - In Comparison and applicability, in the second paragraph, "CSH is an obvious..." Please define and spell out CSH.

 Response: CSH is mentioned and defined earlier in the manuscript in the last paragraph in “Cases of breast cancer in trans men”. However, we spelled it out again. 

#4- In the Introduction, it is stated that the highest incidence of breast cancer is found in women aged 70 - 74 years. However, after checking the reference, this is very likely true only for Sweden; please verify that it is broadly applicable or clarify that this is only for Sweden.

 Response: Yes, great point. We´ve changed the sentence to “For the Swedish population the highest incidence is found in women aged 70 to 74 years.3”

#5 - It is curious that only the first author screened all articles, read all text, and gathered all statistics. There was no double-check by the other authors (what exactly did they do?) and is a risk of bias. It is also curious that no protocol was established, and the review has not been registered anywhere.

 Response: The inclusion/exclusion process were being conducted by the first author with supervision from the third author. Inclusion/exclusion criteria were clearly stated before initiation of the search. The population were also clearly defined as trans men having had some form of gender dysphoric intervention made. These criteria were strictly followed and no evident issues with inclusion/exclusion were found. In those cases, the third author would be available for discussion. The process was especially straight forward with regards to title and abstract screening. For transparency we provide all excluded articles from the full-text screening in the S1 table, with reasons for exclusions. 

Whilst no formal protocol was produced as a separate document before initiating the study, the search was clearly defined beforehand using the PICO approach. The problem/population = breast cancer in trans men. The intervention/risk = GD intervention. Comparison = cis women/trans men with no GD intervention. Outcome = breast cancer case. It is unfortunate that the study was not registered prospectively, as this could help to avoid future “unplanned duplication reviews”. However, we do not consider this fact to introduce any major bias since we´ve made effort to report on the search terms/methodology as transparent as possible to facilitate reproducibility.

#6- Although the author is undoubtedly correct that there is more breast tissue remaining after CCM than after prophylactic mastectomy, it should still somewhat lower the risk of malignancy in trans men than in cis women at normal risk.

 Response: Yes, the CCM alone is probably risk-reducing and we cite Boice (reference nr 63) who shows that a breast reduction in cis women leads to a lower incidence in breast cancer. This is most likely true also for trans men. However, to what degree is impossible to say given the available evidence. 

Reviewer #2: In the references section, this reviewer noticed two typographical errors:

#1 Line 602, San Francisco is misspelled ("San fransisco" is incorrect)

 Response: Corrected.

#2 Line 604, the second author on the cited document is Goldberg, J (not Joshua G)

 Response: Yes, thanks for noticing. Corrected.

---

## [Decision Letter · Decision Letter 1]

9 Feb 2024

Aspects to consider regarding breast cancer risk in trans men: a systematic review and risk management approach.

PONE-D-23-41281R1

Dear Dr. Wahlström,

We’re pleased to inform you that your manuscript has been judged scientifically suitable for publication and will be formally accepted for publication once it meets all outstanding technical requirements.

Kind regards,

Daniele Ugo Tari, M.D.

Academic Editor

PLOS ONE

Additional Editor Comments (optional):

Reviewers' comments:

Reviewer's Responses to Questions

**Comments to the Author**

1. If the authors have adequately addressed your comments raised in a previous round of review and you feel that this manuscript is now acceptable for publication, you may indicate that here to bypass the “Comments to the Author” section, enter your conflict of interest statement in the “Confidential to Editor” section, and submit your "Accept" recommendation.

Reviewer #1: All comments have been addressed

2. Is the manuscript technically sound, and do the data support the conclusions?

Reviewer #1: (No Response)

3. Has the statistical analysis been performed appropriately and rigorously? 

Reviewer #1: (No Response)

4. Have the authors made all data underlying the findings in their manuscript fully available?

Reviewer #1: (No Response)

5. Is the manuscript presented in an intelligible fashion and written in standard English?

Reviewer #1: (No Response)

6. Review Comments to the Author

Reviewer #1: (No Response)

7. PLOS authors have the option to publish the peer review history of their article (what does this mean?). If published, this will include your full peer review and any attached files.

Reviewer #1: No

---

## [Editor Report · Acceptance letter]

26 Feb 2024

PONE-D-23-41281R1 

PLOS ONE

Dear Dr. Wahlström, 

I'm pleased to inform you that your manuscript has been deemed suitable for publication in PLOS ONE. Congratulations! Your manuscript is now being handed over to our production team.

Kind regards, 

on behalf of

Dr. Daniele Ugo Tari 

Academic Editor

PLOS ONE